# Pantothenate and L-Carnitine Supplementation Improves Pathological Alterations in Cellular Models of KAT6A Syndrome

**DOI:** 10.3390/genes13122300

**Published:** 2022-12-06

**Authors:** Manuel Munuera-Cabeza, Mónica Álvarez-Córdoba, Juan M. Suárez-Rivero, Suleva Povea-Cabello, Irene Villalón-García, Marta Talaverón-Rey, Alejandra Suárez-Carrillo, Diana Reche-López, Paula Cilleros-Holgado, Rocío Piñero-Pérez, José A. Sánchez-Alcázar

**Affiliations:** Centro Andaluz de Biología del Desarrollo (CABD-CSIC), Universidad Pablo de Olavide, 41013 Sevilla, Spain

**Keywords:** intellectual disability, KAT6A syndrome, lysine acetyltransferase 6 A, pantothenate, L-carnitine, histone acetylation

## Abstract

Mutations in several genes involved in the epigenetic regulation of gene expression have been considered risk alterations to different intellectual disability (ID) syndromes associated with features of autism spectrum disorder (ASD). Among them are the pathogenic variants of the lysine-acetyltransferase 6A (*KAT6A*) gene, which causes KAT6A syndrome. The KAT6A enzyme participates in a wide range of critical cellular functions, such as chromatin remodeling, gene expression, protein synthesis, cell metabolism, and replication. In this manuscript, we examined the pathophysiological alterations in fibroblasts derived from three patients harboring KAT6A mutations. We addressed survival in a stress medium, histone acetylation, protein expression patterns, and transcriptome analysis, as well as cell bioenergetics. In addition, we evaluated the therapeutic effectiveness of epigenetic modulators and mitochondrial boosting agents, such as pantothenate and L-carnitine, in correcting the mutant phenotype. Pantothenate and L-carnitine treatment increased histone acetylation and partially corrected protein and transcriptomic expression patterns in mutant KAT6A cells. Furthermore, the cell bioenergetics of mutant cells was significantly improved. Our results suggest that pantothenate and L-carnitine can significantly improve the mutant phenotype in cellular models of KAT6A syndrome.

## 1. Introduction

Currently, with the development of next-generation sequencing (NGS) techniques, numerous pathological variants in many genes participating in the epigenetic regulation of gene transcription have been considered risk genes for several intellectual disability (ID) syndromes that are often associated with autism spectrum disorders (ASDs) [1,2,3,4,5]. Among them are de novo mutations of the lysine-acetyltransferase 6A gene (*KAT6A*; also known as *MYST3* and *MOZ*; MIM *601408) that cause KAT6A syndrome (Arboleda-Tham Syndrome, autosomal dominant mental retardation 32; MIM # 616268) [6]. Autism and autistic features have been reported in approximately 25% of newly reported cases of KAT6A syndrome [1].

The *KAT6A* gene, located in chromosome 8p11.21, encodes a lysine-acetyltransferase 6A (KAT6A). KAT6A belongs to the MYST (named for members MOZ, Ybf2/Sas3, Sas2, and Tip60) family of histone acetyltransferases that are characterized by the presence of a well-conserved MYST sequence containing an acetyl-CoA binding domain and a zinc finger motif [7]. Proteins of the MYST family (KAT6A, KAT6B, KAT5, and KAT7) participate in many essential cellular processes, such as chromatin remodeling, regulation of gene expression, protein synthesis, metabolic pathways, and cell division [8]. KAT6A functions in a multisubunit complex with three other proteins: bromodomain and PHD finger 1/2/3 (BRPF1/2/3), Inhibitor of growth family member 5 (ING5), and human Esa1-associated factor 6 (hEAF6) [9]. These proteins make a complex to acetylate lysine residues on histone H3 tails, that way regulating gene expression patterns and promoting diverse developmental programs.

The *KAT6A* gene was identified as a common locus of chromosomal translocations associated with acute myeloid leukemia (AML) [10]. The KAT6A enzyme catalyzes the transfer of acetyl groups to lysine-9 residues in histone H3 (H3K9), playing a critical role in the regulation of gene expression. KAT6A is also involved in the acetylation and regulation of the tumor suppressor p53, a multifactorial protein that is able to control cell cycle progression, DNA integrity, and the survival of the cells exposed to DNA damaging agents [11]. Moreover, KAT6A, through its C-terminal domain, has the ability to bind and regulate several transcription factors, such as Runx1 and Runx2 [12].

Most KAT6A mutations are autosomal-dominant loss-of-function variants, including splicing and stop-codon mutations. Recently, single-base pair substitution affecting functional domains has also been identified [1,13].

Many clinical features in KAT6A syndrome have a variable penetrance (the spectrum of clinical signs and symptoms that manifest in individuals with the same genetic condition). The basic pathological characteristics are microcephaly, ID, speech delay, autism, gastrointestinal complications, and cardiac alterations [1].

There are diverse experimental models to investigate KAT6A mutations, such as the KAT6A knockout mouse that resulted in embryonic lethality due to a failure of hematopoiesis [14]. A knock-in pathogenic variant that eliminates KAT6A’s acetyltransferase function in embryonic stem cells and mouse lines showed proliferation defects, decreased body weight, and decreased life span [14]. Tissue- or cell-specific knockout has shown that KAT6A regulates developmental programs involved in hematopoiesis, skeletogenesis, and thymic and splenic function [14,15,16]. Further studies demonstrated that KAT6A-mediated acetylation induces the generation of memory B-cells and the CD8 T-cell response to viral infection [17,18]. In addition, transcriptomic profiles of human patient-derived fibroblast cell lines harboring heterozygous KAT6A truncating mutations demonstrated changes in the expression of p53-related genes [3].

At present, fibroblast cell cultures generated from patient skin biopsies are useful biological models for exploring the molecular alterations and the response of particular mutations to specific treatments [19]. Thus, cellular and molecular studies of fibroblasts derived from patients with neurological and neurodevelopmental diseases have provided a great deal of useful information on the molecular mechanisms of these disorders [20,21,22,23,24]. The justification for this approach is based on the assumption that although neurological and neurodevelopmental genetic disorders are primarily located within the central nervous system (SNC), patient-derived fibroblasts harbor the particular mutation and can mimic many of the pathological defects observed within the SNC.

In this manuscript, we evaluated the expression levels of proteins involved in acetylation-deacetylation reactions, coenzyme A (CoA) metabolism, mitochondrial proteins, iron metabolism, and antioxidant enzymes in cellular models derived from three KAT6A patients. In addition, we evaluated the effect of epigenetic modulators and mitochondrial boosting agents, such as pantothenate and L-carnitine, on transcriptome profiles, protein expression levels, and cell bioenergetics.

## 2. Materials and Methods

### 2.1. Reagents

Sodium pantothenate (17288) was purchased from Cayman Chemicals (Michigan). Anti-divalent metal transporter 1 (DMT1) (ABS983), fetal bovine serum (FBS) (F7524), Oligomycin A (75351-5MG), and Prussian blue (03899-25g) were purchased from Sigma Chemical Co. (St. Louis, MO). DAPI (sc-3598), L-carnitine (sc-205727), Carbonyl cyanide 4-(trifluoromethoxy) phenylhydrazone (FCCP) (sc-203578), Antimycin A (sc-202467A), Paraformaldehyde (sc-253236), Rotenone (sc-203242), anti-superoxide dismutase 1 (SOD1) (sc-101523), and anti-ferritin light chain (sc-74513), were purchased from Santa Cruz Biotechnology (Santa Cruz, CA). A Histone H3 Total Acetylation Detection Fast Kit (Colorimetric) (ab115124), an NAD/NADH Assay Kit (ab65348), anti-nuclear receptor coactivator 4 (NCOA4/ARA70) (ab86707), anti-sirtuin 1 (SIRT1) (ab110304), anti-voltage-dependent anion channel 1 (VDAC1) (ab14734), anti-ATP5F1A (ab14748), anti-nitrogen fixation 1 (NFS1) (ab58623), superoxide dismutase 2 (SOD2) (ab68155), anti-cytochrome C oxidase subunit 4 (COX4) (ab14744), anti-nicotinamide phosphoribosyltransferase 1 (NAMPT1) (ab236874), and anti-mitochondrial ferritin (ab124889) were purchased from Abcam (Cambridge, UK). Dulbecco’s Modified Eagle’s Medium (DMEM) (10524684), Dulbecco’s Modified Eagle’s Medium without glucose (DMEM) (11966025), Penicillin-Streptomycin (11548876), MitoTracker™ CMXRos FM (M7512), anti-pantothenate kinase 2 (PANK2) (CF501355), anti-lysine acetyltransferase 6A (KAT6A/MOZ) (PA568046), anti-glutathione peroxidase 4 (GPX4) (MA5-32827), anti-mitochondrial acyl carrier protein (mtACP) (PA5-30099), anti-sirtuin 3 (SIRT3) (PA5-13222), anti-aminoadipate semialdehyde dehydrogenase phosphopantetheinyl transferase (AASDHPPT) (PA5-39222), anti-NADH: ubiquinone oxidoreductase subunit A9 (NDUFA9) (459100), anti-iron-sulfur cluster assembly enzyme (ISCU) (MA5-26595), anti-mitoferrin 2 (PA5-42498), anti-transferrin receptor (TfR) (13-6800), and trypsin (15090-046) were purchased from Thermo-Fisher Scientific (Waltham, MA). Anti-lipoic acid (437695-100UL) was acquired from Merck Millipore (Darmstadt, Germany). Anti-lysine acetylated (9441) was purchased from Cell Signaling Technology (Danvers, MA). Anti-β actin (MBS448085), anti-mitochondrially encoded NADH: ubiquinone oxidoreductase core subunit 6 (MT-ND6) (MBS8518686), and anti-H3K9/K14 acetylated (MBS840797) were purchased from MyBioSource (San Diego, CA). A cocktail of protease inhibitors (complete cocktail) (5892970001) was purchased from Boehringer Mannheim (Indianapolis, IN). Anti-Frataxin (FXN) (LS-C755462) was purchased from LS Bio (Seattle, WA). A Takara PCR Mycoplasma Detection Set (6691) was purchased from Clontech. Plasmocure™-Mycoplasma Elimination Reagent (ant-pc) was purchased from InvivoGen. Agilent Seahorse XF Base Medium (102353-100) and Seahorse XFe24 FluxPak (102340-100) were purchased from Agilent (Santa Clara, CA, USA).

### 2.2. Ethical Statements

Approval of the ethical committee of the Hospital Universitario Virgen Macarena y Virgen de Rocío de Sevilla (Spain) was obtained, according to the principles of the Declaration of Helsinki and the International Conference on Harmonization and Good Clinical Practice Guidelines.

### 2.3. Cell Culture

We used primary skin fibroblast from two healthy subjects (Control 1 and 2), purchased from the American Type Culture Collection (ATCC), and from three patients harboring KAT6A mutations. One of the patients (P1) has a heterozygous mutation c. [3427–3428 ins TA] that causes a frameshift (p. Ser1143Leu), resulting in a premature stop codon that is predicted to be pathological by prediction tools such as PolyPhen2 [25]. The second patient (P2) is a heterozygous carrier of a change in position c. [1075 G > A] (p. Gly359Ser), resulting in an amino acid change in the acetyltransferases domain that likely causes a loss of function. The third patient (P3) carries a heterozygous mutation c. [3385 C > T] that causes a premature stop codon (p. Arg1129*). Fibroblasts were grown in Dulbecco’s Modified Eagle’s Medium (DMEM-Sigma) supplemented with 10% fetal bovine serum (FBS-Sigma), 100 mg/mL streptomycin, 100 U/mL penicillin, and 4 mM l-glutamine (Sigma). All the experiments were performed with fibroblast cell cultures with a passage number <8. Cell cultures were cleaned from mycoplasma with Plasmocure™-Mycoplasma Elimination Reagent and tested using the Takara PCR Mycoplasma Detection Set.

### 2.4. Immunoblotting

Western blotting was performed using standard protocols [26]. Membranes were incubated with primary antibodies diluted between 1:500 and 1:1000 overnight. Then, the membranes were incubated with the corresponding secondary antibody coupled to horseradish peroxidase at a 1:10,000 dilution. Protein bands were recognized using the Immun Star HRP substrate kit (Biorad Laboratories Inc., Hercules, CA, USA).

Expression levels of actin were examined to assess whether the samples were uniformly loaded across the gel. Membranes were re-probed with different antibodies if the molecular weight of proteins did not interfere. When proteins had different molecular weights, membranes were also cut and incubated with specific antibodies. Three biological replicates were used per immunoblot.

### 2.5. Drug Screening

Drug screenings were performed in nutritional stress medium, with galactose as the main carbon source. This medium deprives cells from glycolysis as an energy source and thereby causes them to rely exclusively in the mitochondrial electron transport chain for adenosine triphosphate (ATP) production [27,28]. In addition, mitochondrial ATP production was slightly impaired with oligomycin at a low concentration (0.5 nM). In this medium, mutant KAT6A fibroblasts were unable to survive.

Fibroblasts were cultured in DMEM and treated for 15 days with several compounds at different concentrations. A nutritional stress medium was prepared with DMEM without glucose, 10 mM galactose, 0.5 nM oligomycin, 1% of antibiotic solution, and 10% FBS. First, fibroblasts were seeded in 24-well plates in DMEM. After 24 h, cells were treated again for 72 h with the same compound at the same concentration. Then, the cell culture medium was removed and cells were cultured in stress medium (Time 0). Thereafter, the treatments were re-applied at the same concentration. The 72 h endpoint was selected because cells showed a significant cell proliferation/death at this time. Cell viability was tested by live cell imaging counting and trypan blue 0.2% staining. Cell quantification was performed using the BioTek^TM^ Cytation^TM^ 1 Cell Imaging Multi-Mode Reader (BioTek, Winooski, VT, USA). Each drug screening was performed in three biological replicates.

### 2.6. Immunofluorescence Microscopy

Immunofluorescence studies were performed using a protocol previously described by our research group [29]. Cells were cultured on 1 mm width (Goldseal No.1) glass coverslips for 24 to 48 h in DMEM with 20% FBS. Cells were washed once with phosphate-buffered saline (PBS), fixed in 3,8% paraformaldehyde in 0,1% of saponin for 5 min. For immunostaining, glass coverslips were incubated with primary antibodies diluted 1:100 in PBS for 1 to 2 h at 37 °C in a humidified chamber. The excess antibodies were removed by washing with PBS (three times during 5 min). Then, secondary antibodies diluted in 1:1000 in PBS were added and incubated for 1 h at 37 °C. Coverslips were then rinsed with PBS for 3 min, incubated for 1 min with PBS containing Hoechst 33,342 (1 µg/mL), and washed with PBS (3 times, 5 min). Finally, coverslips were mounted onto microscope slides using Vectashield Mounting Medium (Vector Laboratories, Burlingame, CA, USA) and analyzed using an upright fluorescence microscope (Leica DMRE, Leica Microsystems GmbH, Wetzlar, Germany). Each immunofluorescence assay was performed in three biological replicates.

### 2.7. RNAseq

Fibroblasts were cultured until confluence. Then, cells were tested for mycoplasma by PCR and cellular pellets were obtained. RNA was extracted and purified using RNeasy Mini Kit (QIAGEN, Hilden, Germany). DNase digestion was performed with the RNase-Free DNase Set (QIAGEN, Hilden, Germany). RNAseq was performed by Microomics Systems S.L. (Barcelona, Spain).

Libraries were prepared using the TruSeq stranded mRNA Library Prep (96 samples ref. 20020595 or 48 samples ref. 20020594), according to the manufacturer’s protocol, to convert total RNA into a library of template molecules of known strand origin and suitable for subsequent cluster generation and DNA sequencing.

Briefly, 1000 or 500 ng of total RNA were used for poly(A)-mRNA selection using poly-T oligo attached magnetic beads with two rounds of purification. During the second elution of the poly-A RNA, the RNA was fragmented under elevated temperature and primed with random hexamers for cDNA synthesis. Then, the cleaved RNA fragments were copied into first-strand cDNA using reverse transcriptase (SuperScript II, ref. 18064-014, Invitrogen) and random primers. Then, second-strand cDNA was synthesized, removing the RNA template and synthesizing a replacement strand, incorporating dUTP in place of dTTP to generate ds cDNA using DNA Polymerase I and RNase H.

These cDNA fragments then received the addition of a single ‘A’ base to the 3′ ends of the blunt fragments to prevent them from ligating to one another during the adapter ligation. A corresponding single T nucleotide on the 3′ end of the adapter provided a complementary overhang for ligating the adapter to the fragments. Subsequent ligation of the multiple indexing adapter to the ends of the ds cDNA was performed. Finally, PCR selectively enriched those DNA fragments that had adapter molecules on both ends. The PCR was performed with a PCR primer cocktail that annealed to the ends of the adapters.

Final libraries were analyzed using Bioanalyzer DNA 1000 or Fragment Analyzer Standard Sensitivity (ref: 5067-1504 or ref: DNF-473, Agilent) to estimate the quantity and validate the size distribution. The libraries were then quantified by qPCR using the KAPA Library Quantification Kit KK4835 (REF. 07960204001, Roche) prior to the amplification with Illumina’s cBot. The libraries were sequenced with 125-bp paired-end reads on Illumina’s HiSeq2500.

The sequencing coverage was around 25 million reads. Raw demultiplexed forward and reverse reads were processed using the following steps: reads of quality control of RNA with FastQC v.0.11.8 with a Phred score (Q) greater than 30 (Appendix A) and pre-processing with Trimmomatic v0.39 [30]. The primary processing was carried out using the following steps: alignment to genome reference Homo_sapiens.CRGh38.102 from Ensembl using Bowtie2 v 2.3.5.1 [31], alignment quality control with qualjmap v.2.2.2 [32], a counts table with featureCounts v1.6.4 [33], and differential expression analysis with DeSeq2 v1.24.0 [34]. Genes differentially expressed had a *p*-value < 0.05, a log2FoldChange negative value between -13.89 and -0.048, and a positive value between 0.043 and 13.24. Enrichment scores were calculated using the methods of Merico et al. [35].

### 2.8. Cell Fractioning

Cells were cultured until confluence; cell pellets were homogenized using a fractionation buffer that contained 250 mM sucrose, 10 mM Tris, 1 mM ethylene diamine tetra acetate (EDTA), and a proteases inhibitors cocktail, pH 7.4. Then, cell homogenates were passed through a 25-gauge needle 8 times using a 1 mL syringe. Next, intact cells and nuclei were removed by centrifugation at 1500× *g* for 20 min. Supernatants with intact mitochondria were centrifuged at 12,000× *g* for 10 min (pellet at the bottom is the “mitochondria fraction”; supernatant is the “cytosolic fraction”). Cytosolic fractions were concentrated using Centricon YM-10 devices (Millipore).

### 2.9. Bioenergetics

The mitochondrial respiratory function of control and mutant KAT6A fibroblasts were measured using a Mito stress test assay with an XF24 extracellular flux analyzer (Seahorse Bioscience, Billerica, MA, USA, 102340-100), according to the manufacturer’s instructions and previous studies [36,37]. Fibroblasts were seeded at a density of 1.5 × 10^4^ cells/well with 500 µL of growth medium (DMEM medium containing 20% of FBS) in XF24 cell culture plates and incubated for 24 h at 37 °C with 5% of CO_2_. Subsequently, growth medium was removed from the wells, leaving on them only 50 μL medium. Then, cells were washed twice with 500 μL of pre-warmed assay XF base medium (102353-100) supplemented with 10 mM glucose (103577-100), 1 mM glutamine (103579-100), and 1 mM sodium pyruvate (103578-100); pH 7.4); eventually, 450 μL of assay medium (500 μL final) were added. Cells were incubated at 37 °C without CO_2_ for 1 h to allow pre-equilibrating with the assay medium. Mitochondrial functionality was evaluated by the sequential injection of four compounds affecting bioenergetics. The final concentrations of the injected reagents were 1 μM oligomycin, 2 μM carbonyl cyanide 4-(trifluoromethoxy) phenylhydrazone (FCCP), and 1 and 2.5 μM rotenone/antimycin A. Optimal concentrations of inhibitors and uncouplers, as well as the cells’ seeding density, were previously determined. A minimum of five replicates per treatment were used in each experiment. Next, basal respiration, maximal respiration, spare respiratory capacity, and ATP production were quantified.

### 2.10. NAD^+^/NADH Levels

Intracellular nicotinamide adenine dinucleotide (NAD^+^/NADH) levels were assessed using the NAD^+^/NADH Colorimetric Assay Kit (Abcam, Hercules, CA, USA, ab65348). Absorbance was measured using a POLARstar Omega plate reader (BMG Labtech, Offenburg, Germany). Each assay was performed in three biological replicates.

### 2.11. Histone H3 Total Acetylation

Histone H3 total acetylation levels in cellular pellets were assessed by the Histone H3 Total Acetylation Colorimetric Detection Fast Kit (Abcam, Hercules, CA, USA, ab115124). Absorbance was measured using a POLARstar Omega plate reader (BMG Labtech, Offenburg, Germany). Each assay was performed in three biological replicates. 

### 2.12. Statistical Analyses

Statistical analyses were performed as previously described [38]. Without any distributional assumption, we used non-parametric statistics when the number of events was small (n < 30) [39]. In this case, multiple groups were compared using a Kruskal–Walli test. In the case of only two groups, they were compared using the method of the Mann–Whitney test. On the other hand, when the events were higher (n > 30), we used parametric tests. Multiple groups were compared using a one-way ANOVA. After this comparison, we applied a Bonferroni post hoc test to look for significant differences between groups. When we had two groups, they were compared by applying a Student’s t-test with a Welch’s correction. Statistical analysis was performed using GraphPad Prism 7.0 (GraphPad Software, San Diego, CA, USA). The data were reported as representative of at least three independent experiments; *p*-values of less than 0.05 were considered significant.

## 3. Results

### 3.1. Protein Expression Levels in KAT6A Fibroblasts

#### 3.1.1. Expression Levels of the KAT6A Enzyme Are Markedly Reduced in Fibroblasts Derived from KAT6A Patients

First, we analyzed the protein expression levels of the mutant KAT6A enzyme in fibroblast cell lines derived from three KAT6A patients and two healthy subjects (C1 and C2). The three patients had heterozygous mutations in the *KAT6A* gene. Patient 1 (P1) had an insertion of two amino acids that resulted in a premature stop codon; patient 2 (P2) had a base change; and patient 3 (P3) had a base change that resulted in a premature stop codon. The KAT6A protein expression levels were markedly reduced in the three patient cell lines (Figure 1A and Appendix A). Curiously, the KAT6A expression levels were higher in P2 fibroblasts than in P1 and P3 fibroblasts, suggesting that the expression levels of the mutant enzyme may depend on the type of mutation. In addition, SIRT1 and SIRT3 (sirtuins 1 and 3), nicotinamide adenine dinucleotide (NAD^+^)-dependent protein deacetylases, and NAMT (Nicotinamide phosphoribosyltransferase), the rate-limiting enzyme in the NAD^+^ salvage pathway, were also downregulated in mutant cells (Figure 1A and Appendix A). These results suggest that KAT6A mutations lead to downregulation of enzymes involved in both acetylation and deacetylation processes.

#### 3.1.2. Expression of Proteins Involved in Coenzyme A (CoA) Metabolism Were Also Affected in Mutant KAT6A Fibroblasts

As the KAT6A enzyme uses acetyl-CoA as a substrate for histones acetylation, we next addressed the expression levels of proteins implicated in CoA metabolism and downstream proteins, such as mtACP (mitochondrial acyl carrier protein), mitochondrial lipoylated proteins, and AASDHPPT (aminoadipate-Semialdehyde Dehydrogenase-Phosphopantetheinyl Transferase), and enzymes involved in the hydrolysis of CoA and the transfer of the 4′-phosphopantetheinyl moiety to mitochondrial proteins such as mtACP [40,41]. The expression levels of pantothenate kinase 2 (PANK2), mtACP, lipoylated PDH (pyruvate dehydrogenase), lipoylated KGDH (α-ketoglutarate dehydrogenase), and AASDHPPT were markedly reduced in mutant KAT6A fibroblasts (Figure 1B and Appendix A). These results suggest that proteins involved in CoA biosynthesis and downstream CoA-dependent pathways are downregulated in mutant KAT6A fibroblasts.

#### 3.1.3. Expression Levels of Mitochondrial Respiratory Chain Proteins Were Affected in KAT6A Mutant Fibroblasts

To address the pathological alterations of KAT6A deficiency in patient-derived fibroblasts, we next examined the expression levels of proteins involved in the mitochondrial respiratory chain. The expression levels of several mitochondrial subunits, such as NDUFA9, COX4, Mt-ND6, and ATP5F1A, were notably reduced in KAT6A fibroblasts (Figure 1C and Appendix A). In contrast, the expression levels of VDAC1, a marker of mitochondrial content of cells [42], were not affected (Appendix A). These results suggest that there is a downregulation of several essential mitochondrial proteins in KAT6A fibroblasts. The decreased levels of essential mitochondrial proteins may lead to mitochondrial dysfunction, increased reactive oxygen species (ROS) production, and reduced energy generation.

#### 3.1.4. Expression Levels of Several Proteins Implicated in Iron Metabolism Were Reduced in Mutant KAT6A Fibroblasts

As PANK2 and mtACP deficiency may alter iron metabolism [40], we then explored the expression levels of several proteins involved in iron handling. As displayed in Figure 1D and Appendix A, the expression levels of several proteins related to this pathway were reduced, such as transferrin receptor (TfR), DMT1, ferritin, mitoferrin 2, mitochondrial ferritin, and NCOA4, which were markedly reduced in mutant KAT6A fibroblasts. In contrast, the expression levels of proteins involved in iron–sulfur clusters’ biosynthesis, such as ISCU (iron–sulfur cluster assembly enzyme), NFS1 (*NFS1* cysteine desulfurase), and FXN (Frataxin), were not affected. In addition, intracellular iron accumulation assessed by Prussian blue staining was not observed in mutant KAT6A fibroblasts (Appendix A).

#### 3.1.5. Expression Levels of Antioxidant Enzymes Were Also Reduced in Mutant KAT6A Fibroblasts

As mitochondrial dysfunction can increase oxidative stress, we also addressed the protein expression levels of antioxidant enzymes. The protein expression levels of SOD1, SOD2, and GPX4 were markedly reduced in mutant KAT6A fibroblasts (Figure 1E and Appendix A). These results indicate that the enzymatic antioxidant system is downregulated in mutant KAT6A fibroblasts.

### 3.2. Effect of Pantothenate on KAT6A Fibroblasts

#### 3.2.1. Pantothenate and L-Carnitine Supplementation Enhance the Survival of the Mutant KAT6A Fibroblasts in Nutritional Stress Medium

As mitochondrial dysfunction can be a critical pathological feature in mutant KAT6A fibroblasts, we developed a screening protocol in the nutritional stress medium to force mitochondrial function and induce cell death in mutant fibroblasts. In these conditions, cell survival rescue by pharmacological compounds is an interesting approach for the identification of supplements capable of correcting the mutant phenotype. Thus, the control and mutant P1 fibroblasts were cultured for 15 days on a glucose-rich DMEM medium with or without supplements at different concentrations. Then, the medium was replaced by a galactose medium with 0.5 nM of oligomycin. As expected, no differences could be observed on the cell proliferation in the control fibroblasts (Figure 2A,B). In contrast, KAT6A fibroblasts did not survive in the stress medium (Figure 2C,D and Appendix A). Curiously, supplementation with 0.8 µM of pantothenate, a CoA metabolism activator, or 0.8 µM of L-carnitine, a mitochondrial boosting agent, enabled the survival of mutant KAT6A fibroblasts in the stress medium (Figure 2E–H and Appendix A), although the cell proliferation was slower than it was in the normal medium. Interestingly, the combined treatment with pantothenate and L-carnitine showed a synergic positive effect in cell survival. In addition, KAT6A cells survived at lower concentrations of pantothenate and L-carnitine (Figure 2I,J and Appendix A).

The lowest concentrations of pantothenate and L-carnitine necessary for cell survival in the stress medium varied among the different KAT6A cell lines, suggesting that KAT6A mutations may respond differently to pantothenate and L-carnitine concentrations (Appendix A). We found that patient 1’s and patient 3’s cells survived at 0.7 µM of pantothenate and 0.7 µM of L-carnitine, while patient 2’s cells survived at 0.4 µM of pantothenate and 0.4 µM of L-carnitine.

#### 3.2.2. Pantothenate and L-Carnitine Supplementation Partially Correct Protein Expression Patterns in Mutant KAT6A Cell Lines

Next, we assessed the positive effects of pantothenate and L-carnitine supplementation on the expression levels of the mutant KAT6A enzyme, SIRT1, SIRT3, NAMPT, mitochondrial proteins (Mt-ND6 and NDUFA9), CoA metabolism-related proteins (PANK2, mtACP, lipoylated PDH, and lipoylated KGDH), and antioxidant enzymes (SOD1, SOD2, and GPX4). The expression levels of these proteins were markedly downregulated in the three KAT6A cell lines, although to different extents, due to the broad diversity in genetic backgrounds and the types of mutations. Interestingly, treatment with pantothenate and L-carnitine enhanced the expression levels of all downregulated proteins (Figure 3 and Appendix A). Different concentrations of pantothenate and L-carnitine were used in the mutant cell lines, according to the minimum concentration required for survival in the screening assay in the stress medium. Patient 1 and patient 3 were treated with 0.7 µM L-carnitine and 0.7 µM pantothenate (Figure 3A,C and Appendix A), while patient 2 was treated with 0.4 µM L-carnitine and 0.4 µM pantothenate (Figure 3B and Appendix A).

#### 3.2.3. Pantothenate and L-Carnitine Supplementation Increases Histones Acetylation in KAT6A Cells

As the KAT6A enzyme acetylates lysine 9 and lysine 14 of histone H3, we next assessed the efficacy of pantothenate and L-carnitine in improving acetylation activity. For this purpose, the total acetylations of histone H3 were determined in the Control and mutant KAT6A P1 fibroblasts by immunofluorescence and a colorimetric assay in nuclear fractions. Histone acetylation of mutant KAT6A fibroblasts was significantly reduced in mutant KAT 6A fibroblasts (Figure 4A,B and Appendix A). Interestingly, the supplementation of mutant cells with pantothenate and L-carnitine induced a marked increase in histone acetylation, reaching levels similar to those of control cells (Figure 4A,B and Appendix A). These findings suggest that histone acetylation deficiency in KAT6A fibroblasts was corrected by pantothenate and L-carnitine supplementation. The positive effect of pantothenate and L-carnitine on histone acetylation was also confirmed in P2 and P3 mutant cell lines (Appendix A).

Next, to address the effect of pantothenate and L-carnitine on sirtuins cofactors involved in histone deacetylation reactions by SIRTs, total NAD (NADt), NAD^+^, NADH levels and the NAD^+^/NADH ratio were determined in the P1 mutant cells (Figure 5A–D). Our results showed that NADt, NAD^+^, and NADH content and the NAD^+^/NADH ratio were significantly reduced in KAT6A mutant cells, and that pantothenate and L-carnitine treatment was able to correct their intracellular levels. These findings suggest that pantothenate and L-carnitine treatment also corrects the content of NAD^+^, which is an essential cofactor for sirtuins’ function and histone deacetylation reactions.

#### 3.2.4. Pantothenate and L-Carnitine Supplementation Improves Cell Bioenergetics in KAT6A Mutant Fibroblasts

To test the efficacy of pantothenate and L-carnitine treatment in improving mitochondrial activity, we assessed mitochondrial membrane potential by Mitotracker staining and cell bioenergetics using the XF Cell Mito Stress assay. As expected, mitochondrial bioenergetics was altered in mutant KAT6A fibroblasts, presenting a general decrease in mitochondrial membrane potential (Figure 4 and Appendix A) and respiratory parameters (Figure 6A,B). Confirming the positive effect previously observed, 0.7 µM pantothenate and 0.7 µM L-carnitine supplementation significantly restored mitochondrial membrane potential (Figure 4 and Appendix A), mitochondrial maximal respiration, and spare respiratory capacity in mutant KAT6A fibroblasts (Figure 6A,B). These results are consistent with the recovery of mitochondrial protein expression levels under supplementation with pantothenate and L-carnitine (Figure 3) and the recovery of mitochondrial membrane potential (Figure 4A and Appendix A). The positive effect of pantothenate and L-carnitine on mitochondrial membrane potential and cell bioenergetics was also confirmed in the P2 and P3 mutant cell lines (Appendix A).

#### 3.2.5. Pantothenate and L-Carnitine Treatment Highly Modifies the Transcriptome

To analyze the pathological effects of KAT6A mutations and to assess the effect of 0.7 µM pantothenate and 0.7 µM L-carnitine supplementation on gene expression, we next performed an RNA-seq (RNA sequencing) analysis on the control and mutant KAT6A P1 fibroblasts, with and without pantothenate/carnitine treatment. The RNA-seq was meant to provide a resulting expression value for each gene and an average of its expression levels across the different conditions. Among the 60675 expressed genes that were detected, 12719 showed differential expressions in the control and mutant KAT6A fibroblasts, 10026 showed differential expressions in the mutant KAT6A fibroblasts and treated KAT6A mutant fibroblasts, and 11530 showed differential expressions in the control and treated mutant KAT6A fibroblasts Figure 7A,B,D,E,G,H. Moreover, we assessed the effect of pantothenate and L-carnitine treatment on the control fibroblasts. The results showed that only 134 genes were differentially expressed in the control cells after treatment.

Since the human KAT6A protein regulation pathway remains obscure, no databases, such as the Kyoto Encyclopedia of Genes and Genomes (KEGG), are available to study the proteins involved in it. Consequently, biological process ontology (BP) was used to assess which of the differentially expressed genes might be related to KAT6A protein. BP analysis indicated that genes related to acetylation/methylation were decreased and neuronal regulation genes were increased in mutant KAT6A fibroblasts when compared with control fibroblasts (Figure 7C–F). Interestingly, these pathways recovered their normal expression pattern under pantothenate and L-carnitine supplementation (Figure 7F). No significant differences were found between the control and treated mutant KAT6A fibroblasts. These results suggest that there is a recovery on gene expression patterns under pantothenate and L-carnitine supplementation. As we can see in Figure 7I, the PCA (principal component analysis) plot showed that treated mutant KAT6A fibroblast partially recovered the transcriptomic pattern of the control cells.

Next, we analyzed the expression changes of specific genes in treated and untreated conditions. We found that the expression levels of downregulated genes in the mutant KAT6A fibroblasts, such as *KAT6A*, *SIRT1*, *SIRT3*, *NAMPT1*, *Mt-ND6*, *NDUFA9*, *PANK2*, *mtACP*, *PDH* (*E1 subunit α2*), *KGDH* (*E2 subunit*), *SOD1*, *SOD2*, and *GPX4* were significantly restored after pantothenate and L-carnitine treatment (Appendix A). The proteins encoded by these genes are involved in acetylation-deacetylation pathways, CoA metabolism, mitochondria, and antioxidant enzymes, all of which are critical for intracellular processes in embryonic and childhood development.

## 4. Discussion

In this study, we assessed the pathophysiological alterations present in three patient-derived fibroblasts cell lines carrying *KAT6A* mutations. Our results suggest that patient-derived fibroblasts are interesting biological models for recreating the pathological alterations of the disease. In addition, cellular models can facilitate the screening of a large number of compounds and evaluate their positive effect on altered pathways.

Our results demonstrated that KAT6A enzyme expression levels were markedly reduced in fibroblasts derived from KAT6A patients. Furthermore, we showed that deficiency of KAT6A reduced histone H3 acetylation and led to altered gene expression patterns and downregulation of essential proteins in cell metabolism. Screening survival assays in a nutritional stress medium identified two positive compounds, pantothenate and L-carnitine. Interestingly, the supplementation with pantothenate and L-carnitine increased the expression levels of KAT6A mutant enzyme, accompanied by a significant correction of histone acetylation and the recovery of gene expression patterns and expression levels of affected proteins.

KAT6A (MOZ/MYST3) is a MYST family transcriptional coactivator with histone acetyltransferase activity [43]. Transcriptional coactivators work together with other proteins to positively regulate the transcription of certain genes. They form multiprotein complexes that are recruited to specific genomic localizations by DNA-binding transcription factors. Coactivator complexes commonly contain an enzyme subunit with chromatin-modifying activity, such as a histone acetyltransferase [44]. The exchange of a corepressor complex with a coactivator complex having a histone acetyltransferase, such as KAT6A, may be an essential process in inducing gene transcription [45]. Coregulator complexes determine an additional regulation level where transcriptome and, consequently, protein expression and cellular phenotype are modulated [46].

KAT6A is essential for fundamental pathways, such as hematopoietic stem cells, normal B cell development, cell cycle progression, and stem cell self-renewal, among others [16,17,47,48,49]. Thus, dysregulation of these processes due to KAT6A deficiency produces positive and negative expression of proteins involved in these processes, as well as in cellular senescence [50,51].

The acetylation status of histones is produced by the opposing action of histone deacetylases and histone acetyltransferases [52]. Histone acetylation is a critical epigenetic modification that changes chromatin architecture and regulates gene expression by opening or closing the chromatin structure [53]. The decrease of KAT6A protein levels alters the acetylation status of histones and, therefore, the transcription of many essential genes implicated in critical cellular processes [8,11,12,54,55,56]. For all these reasons, the upregulation of the expression levels of KAT6A can be a critical target for restoring histone acetylation and, as consequence, correcting gene expression patterns. Curiously, KAT6A acetylase deficiency was also associated with the downregulation of NAD^+^-dependent deacetylase proteins, such as SIRT1 and SIRT3 (as well as NAMT, which regulates NAD^+^ production and, therefore, SIRTs activity), suggesting that KAT6A mutations secondarily lead to a reduction of deacetylase reactions.

In our work, KAT6A mutations also produced a downregulation in the expression levels of the PANK2 enzyme and CoA-dependent downstream proteins, such as mtACP. Given the essential role of mtACP in lipoic acid biosynthesis [57], mtACP deficiency in mutant cells also led to a decreased lipoylation of key mitochondrial proteins, such as PDH and αKGDH, and caused mitochondrial dysfunction [58]. In agreement with these data, cell bioenergetics was altered in mutant KAT6A cells. Moreover, mitochondrial dysfunction can lead to several deleterious consequences [59], contributing to the development and progression of cell damage in KAT6A syndrome.

It is interesting to note that although PANK2, mtACP, lipoylated PDH, and αKGDH are markedly reduced (Figure 1B), iron is not accumulated in mutant fibroblasts as has been reported in other disorders [40]. Further studies are needed to clarify the absence of iron accumulation in KAT6A mutant cells. 

In KAT6A cellular models, we observed the misregulation of thousands of genes (Figure 7), among which many pathway-specific genes are included. As a consequence, the expression levels of genes encoding essential proteins, such as KAT6A, SIRT1, SIRT3, NAMPT1, Mt-ND6, NDUFA9, PANK2, mtACP, lipoylated PDH, lipoylated KGDH, SOD1, SOD2, and GPX4, are downregulated (Appendix A). These proteins are involved in acetylation–deacetylation pathways, CoA metabolism, mitochondria function, and antioxidant enzymes. These observations are consistent with several reports showing that KAT6A is required for the expression of several genes during development [8,11,12,56].

Interestingly, our results also showed that supplementation with pantothenate and L-carnitine had a positive effect on KAT6A mutant cells associated with the correction of altered protein expression levels, histone acetylation, and cell bioenergetics. The beneficial effect of pantothenate and L-carnitine was also confirmed by RNAseq analysis. Thus, the supplementation with both compounds restored the altered pathological expression of genes related to acetylation/methylation and neuronal regulation. In addition, the expression levels of genes encoding downregulated proteins were restored.

### Pantothenate and L-Carnitine as Epigenetic Modulators

Vitamin B5, or pantothenate, is the precursor of the CoA biosynthetic pathway [60]. The molecule is largely widespread in biology [61] and, in humans, pantothenate deficiency may occur only as a consequence of severe malnutrition. In the cells, CoA synthesis starts with the phosphorylation of pantothenate to 4-phosphopantothenate by PANK. This first reaction represents the major rate-limiting and control step in the entire process of CoA biosynthesis [62]. Pantothenate as a part of CoA forms acetyl-CoA, which is the source of the acetyl group in histone acetylation by the KAT6A enzyme. Therefore, pantothenate supplementation by increasing acetyl-CoA may facilitate the function of the mutant KAT6A enzyme and, therefore, correct the defective histone acetylation in mutant cells. This strategy is based on the idea that a functionally weak/mutant enzyme may work better with higher concentrations of its substrate.

L-carnitine is a ubiquitously occurring trimethylammonium compound that plays a major role in the transport of long-chain fatty acids across the inner mitochondrial membrane and is essential for maintaining normal mitochondrial function and cell metabolism [63,64]. As epigenetic modulator, L-carnitine increases histone acetylation and induces the accumulation of acetylated histones in both normal thymocytes and cancer cells [65]; L-carnitine directly inhibits HDAC I/II (Histone deacetylases I/II) activities and induces lysine-acetylation and histone-acetylation accumulation in vitro [65]. Furthermore, it has been reported that inhibitors of these HDACs boost mRNA and the protein expression of PGC-1α, presumably by promoting the transcription of the *PGC-1a* gene [66]; It is therefore proposed that L-carnitine supplementation may provide a moderate tonic inhibition of type 1 HDACs that supports *PGC1α* transcription and promotes mitochondrial biogenesis.

## 5. Conclusions

In summary, the present work supports the hypothesis that fibroblasts from mutant KAT6A patients are a promising model for the study of the disease’s pathophysiology and the evaluation of potential treatments.

We showed that the decreased expression level of KAT6A protein directly affects histone acetylation affecting critical intracellular processes such as CoA metabolism, iron metabolism, enzymatic antioxidant system, and mitochondrial function. Expression levels of key proteins can be excellent biomarkers to address disease severity and the effectiveness of potential therapies. Interestingly, pantothenate and L-carnitine supplementation increased histone acetylation and significantly rescued protein expression levels and all pathological alterations, including transcriptional patterns and mitochondrial bioenergetics.

## Figures and Tables

**Figure 1 genes-13-02300-f001:**
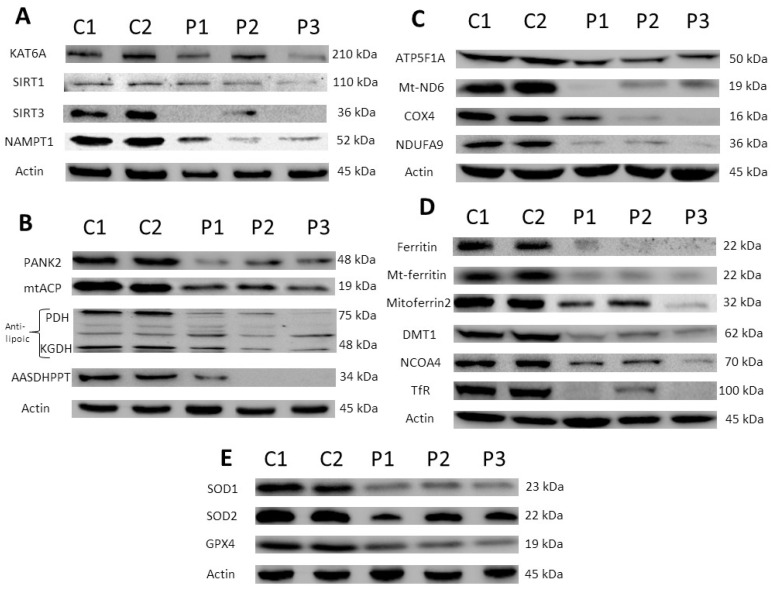
**Protein expression patterns in control and KAT6A mutant fibroblasts.** Protein extracts of Control (C1 and C2) and patient (P1, P2 and P3) cell lines were separated on an SDS polyacrylamide gel and immunostained with primary antibodies. (**A**) Proteins related to acetylation-deacetylation reactions: KAT6A, SIRT1, SIRT3, and NAMPT1; (**B**) proteins related to CoA metabolism: PANK2, mtACP, lipoylated PDH, lipoylated KGDH, and AASDHPPT; (**C**) mitochondrial proteins: ATP Syntase, Mt-NAD6, COX4 subunit, and NDUFA9; (**D**) proteins related to iron metabolism: ferritin, Mt-ferritin, mitoferrin 2, DMT1; NCOA4, and TfR; (**E**) antioxidant enzymes. A representative actin lane is shown, although loading control was additionally checked for every Western blot. Data represent the mean ± SD of three separate experiments. Quantification of protein bands using densitometry is shown in Appendix A.

**Figure 2 genes-13-02300-f002:**
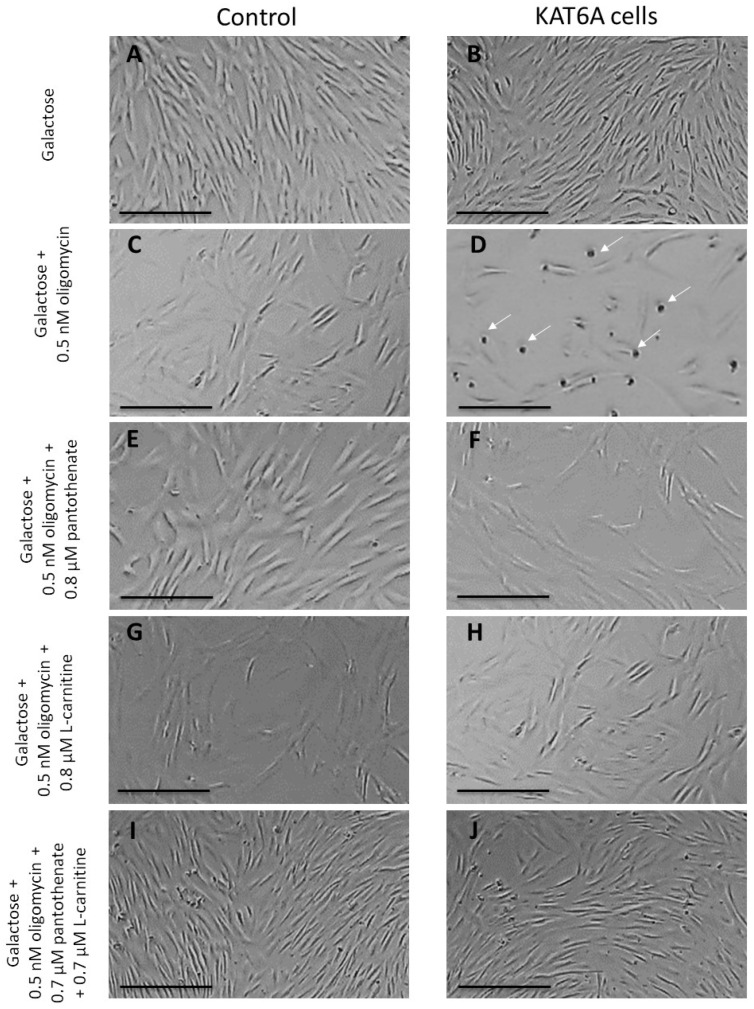
**Pantothenate and L-carnitine supplementation increases cell survival in stress medium**. First, Control (C1) and mutant KAT6A fibroblasts (P1) were cultured in DMEM high glucose. After 3 days, the glucose medium was replaced by a galactose medium with 0.5 nM oligomycin. Images were acquired right after changing the medium and after 72 h of incubation. (**A**,**B**) Control and mutant KAT6A fibroblasts in glucose medium. (**C**,**D**) Control and mutant KAT6A fibroblasts in stress medium. (**E**,**F**) Control and mutant KAT6A fibroblasts (P1) treated with 0.8 µM pantothenate in stress medium. (**G**,**H**) Control and mutant KAT6A fibroblasts treated with 0.8 µM L-carnitine in stress medium. (**I**,**J**) Control and mutant KAT6A fibroblasts (P1) treated with 0.7 µM pantothenate and 0.7 µM L-carnitine in stress medium. Data represent the mean ± SD of three separate experiments. The quantification of cellular proliferation rate is shown in Appendix A. Scale bar = 200 μm. White arrows = dead cells.

**Figure 3 genes-13-02300-f003:**
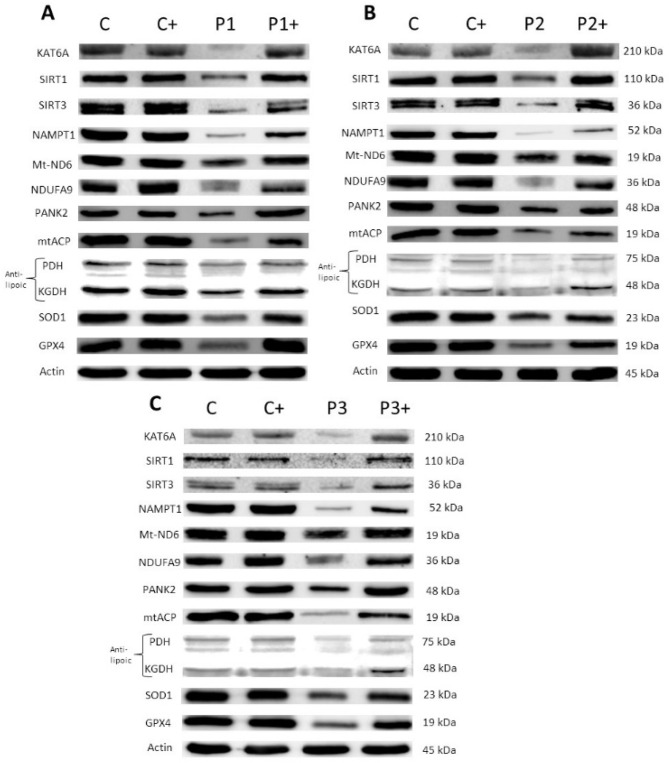
**Pantothenate and L-carnitine supplementation improves protein expression levels in mutant KAT6A fibroblasts**. Control (C) and mutant P1 and P3 fibroblasts (P1 and P3) were treated with 0.7 µM pantothenate and 0.7 µM L-carnitine for 15 days (C+, P1+ and P3+), while Control (C) and P2 fibroblasts (P2) were treated with 0.4 µM pantothenate and 0.4 µM L-carnitine for 15 days (C+ and P2+). Immunoblotting analysis of cellular extracts from Control and P1 fibroblasts (**A**), P2 fibroblasts (**B**), and P3 fibroblasts (**C**). Protein extracts were separated on an SDS polyacrylamide gel (12.5%) and immunostained with antibodies against KAT6A protein, SIRT1, SIRT3, NAMPT1, Mt-ND6, NDUFA9, PANK2, mtACP, lipoic acid (lipoylated PDH and lipoylated KGDH), SOD1, GPX4, and actin. A representative actin band is shown, although loading control was additionally checked for every Western blot. Data represent the mean ± SD of three separate experiments. Protein band densitometry is shown in Appendix A.

**Figure 4 genes-13-02300-f004:**
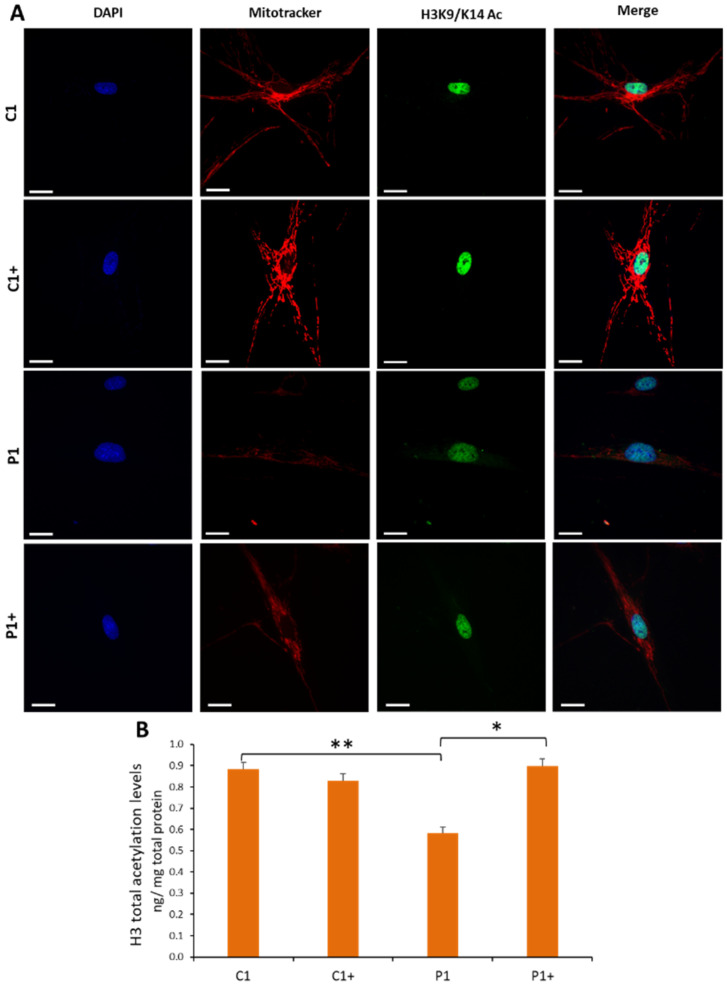
**Pantothenate and L-carnitine treatment increases histone acetylation levels in KAT6A fibroblasts (P1).** Control (C) and mutant P1 fibroblasts (P1) were treated with 0.7 µM pantothenate and 0.7 µM L-carnitine for 15 days (C+ and P1+). (**A**) Control and KAT6A fibroblasts were incubated with Mitotracker CMXRos FM 100 nM for 45 min, then they were fixed and immunostained with anti-H3K9/K14 and examined by fluorescence microscopy. Fifty cells per condition were analyzed. (**B**) Histone H3 total acetylation levels in P1 cellular pellets were assessed by the Histone H3 Total Acetylation Colorimetric Detection Fast Kit (Abcam, Hercules, CA, USA, ab115124) protocol. Data represent the mean ± SD of three separate experiments. Absorbance was measured using a POLARstar Omega plate reader (BMG Labtech, Offenburg, Germany). The mitotracker CMX-ROS and H3K9/K14 intensity assessment were performed using FIJI software, as shown in Appendix A. * *p*-value < 0.05 and ** *p*-value < 0.01. Scale bar = 15 μm. C+ and P1+, treated control and P1 cell lines, respectively.

**Figure 5 genes-13-02300-f005:**
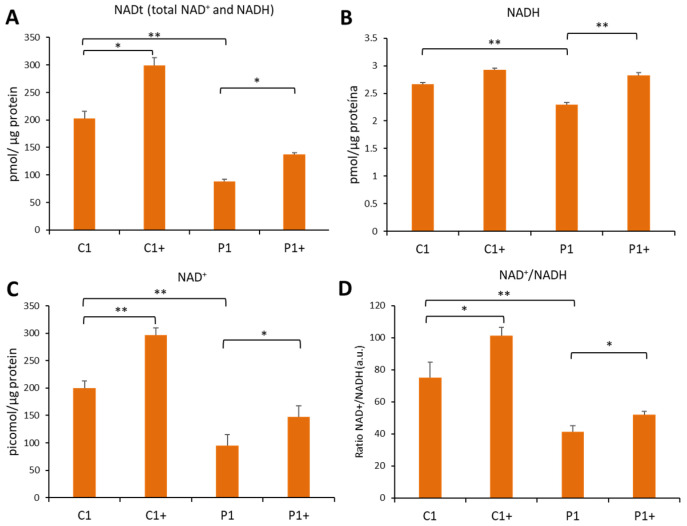
**Pantothenate and L-carnitine supplementation increases cellular NAD^+^/NADH levels in KAT6A mutant fibroblasts (P1)**. Control (C) and KAT6A fibroblasts (P1) were treated for 15 days with 0.7 µM pantothenate and 0.7 µM L-carnitine (C+ and P1+). NAD^+^/NADH assay was performed using the NAD^+^/NADH Assay Kit. NADt (total NAD^+^ and NADH) (**A**) and NADH (**B**) were quantified as described in the Section 2. NAD^+^ was calculated by subtraction (NADt − NADH) (**C**) and NAD^+^/NADH ratio was calculated by the equation ((NADt − NADH)/NADH) (**D**). Data represent the mean ± SD of three separate experiments. * *p*-value < 0.05 and ** *p*-value < 0.01.

**Figure 6 genes-13-02300-f006:**
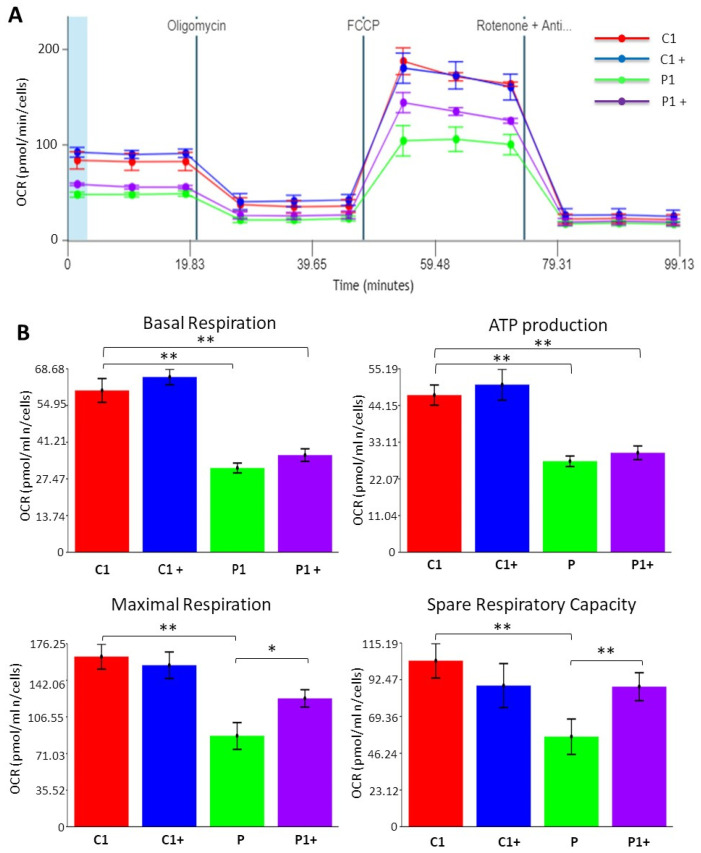
**Pantothenate and L-carnitine supplementation improves cell bioenergetics of mutant KAT6A fibroblasts (P1).** Control (C) and KAT6A fibroblasts (P1) were treated for 15 days with 0.7 µM pantothenate and 0.7 µM L-carnitine (C+ and P1+). (**A**) Mitochondrial respiration profile was measured with a Seahorse XFe24 analyzer. Fibroblasts were treated for 15 days with 0.7 µM pantothenate and 0.7 µM L-carnitine. (**B**) Basal respiration, ATP production, maximal respiration, and spare respiratory capacity were assessed by the Seahorse analytics website. * *p*-value < 0.05 and ** *p*-value < 0.01.

**Figure 7 genes-13-02300-f007:**
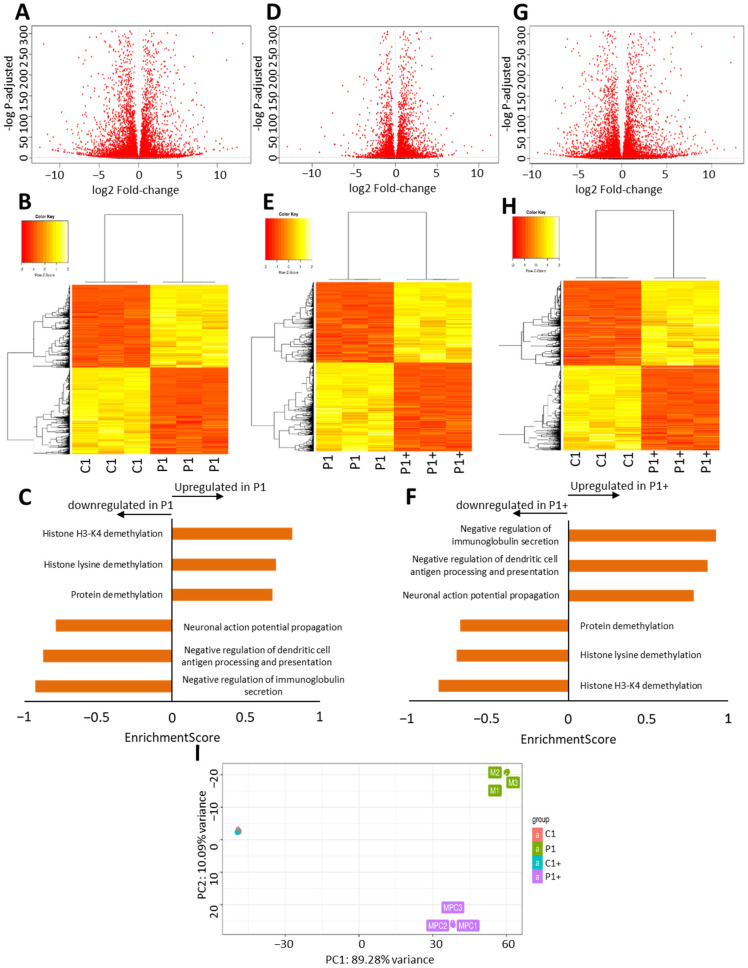
**Pantothenate and L-carnitine supplementation modifies transcriptome of mutant KAT6A fibroblasts**. Control (C) and KAT6A fibroblasts (P1) were treated for 15 days with 0.7 µM pantothenate and 0.7 µM L-carnitine (C+ and P1+). Volcano plot displays the relationship between fold-change and *p*-values (represented as −log p-adjusted, adj) on the differentiation between control and mutant KAT6A fibroblasts (**A**), mutant KAT6A and treated mutant KAT6A fibroblasts (**D**), and control and treated mutant KAT6A fibroblasts (**G**). Genes differentially expressed with P adj < 0.05 are highlighted in red. Heatmap of the relative expression of all differentially expressed genes in the control and mutant KAT6A fibroblasts (**B**), mutant KAT6A and treated mutant KAT6A fibroblasts (**E**), and control and treated mutant KAT6A fibroblasts (**H**). To better interpret RNAseq results, genes were annotated using a functional classification scheme, biological process ontology (BP), which covers gene functions. The results were the comparison between control and mutant KAT6A fibroblasts (**C**) and mutant KAT6A and treated mutant KAT6A fibroblasts (**F**). A PCA (principal component analysis) plot is shown to indicate transcriptomic level differences (**I**).

## Data Availability

The data presented in this study are available in the article and in the Appendix A.

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
