# Peer review of "Pantothenate and L-Carnitine Supplementation Improves Pathological Alterations in Cellular Models of KAT6A Syndrome"

_genes, 2022, doi:10.3390/genes13122300_

Round 1

Reviewer 1 Report

This is an article that states that supplementation of specific metabolites can correct pathological alterations in cellular models of Kat6A . the cell model used is a dermal fibroblast derived from patients with KAT6A syndrome. While the idea is quite interesting, the data to support the very strong claims in the title ‘correction of pathological alterations in cellular models of KAT6A syndrome’. The claims made in writing are not supported by the data shown in the paper, both the conclusions from the data and the lack of biological replicates reported across the paper.

Major Points.

The introduction talks about a lot of items that are not relevant to their message. For example, the first paragraph is about ASD and finding a cure. Furthermore, the reference to autism as associated with ASD is wrong, KAT6A is only considered a class 2 risk gene and not directly causal to ASD. Finally, references 9 and 15 are the same paper—its not clear by it is listed twice. The reproduction should be rewritten to detail the background in the cellular metabolism and the hypothesis they are testing. It is not clear why.how the authors chose  the experimental design and metabolites for testing and what the rational for this.

The title refers “pathological alterations” but its not outline what the authors are are referring to here or in the introduction.

Figure 1, western blots were not properly quantified and its hard to determine whether there was a significant effect for these. The claim of significant effect would require quantitative densitometry measurements of the blot, and testing the relative changes to control across at least 2-3 blots.

Figure 2, is completely uninterpretable. Its not clear what one is supposed to see and therefore any claims here are unjustified and not supported.

Figure 3: the western blot results observed for non-treated P3 different amont the same protein antibody between figure 1 and figure 3.  For example, in Figure 1, for  PDH and KDGH, this fibroblast samples has no expression of these enzymes. However in Figure 3, untreated condition this has quite visible and clear expression. This inconsistence is not explained in the literature.  

Figure 3 & 4 & 5, were only completed with one line and not with the other two. All data should be performed in all three lines. Its impossible to draw any concpusions from a comparison between 1 case and 1 control.  Please use more common notation for p-value assessment rather than the a- and aa notation which is quite confusion.

(page 3, line 101)

One of the main points of the paper suggest that there are new disease modifying therapies poposed from KAT6A. The paper cite is not a peer reviewed paper but an abstract from a small meeting and therefore hard to validate as a to whether a new treatment has truly been proposed.

(Page 3, line 188)

The description of the RNA-analysis is really unclear and have some major analytic problems. First, there is not evidence that there were RNA-seq quality control metrics done. The cells were not mentioned to be tested for mycoplasma and this is quite common in patient-derived lines. The authors talk about RNA-extraction but they do nto give any information about the sequencing metrics (RNA library prep, QC on RNA, read length, coverage or whether the sequencing was paired end or single end). The lack of detail here makes one suspect on data quality. Finally, the authors state that use a p-value of 0.05. Is there any correction for multiple testing here? The lack of details on their RNA-seq process and pipeline further erode credibility on the data quality. x

Minor Points:

This paper includes three patients with three unique mutations. Two are truncating mutations and P2 is a missense mutations. The paper claims this is loss-of-function but there is not proof that this is in fact a true loss-of-function. The authors should dial back their claim on the effects of the mutation.  

Reagents do not have any catalog numbers included and do not include the dilutions tested of the reagents making the work largely non-reproducible.

Typo in page2, line 86 “ very variable penetration” of signs and symptoms  of people with the same condition.

Author Response

REVIEWER 1

Comments and Suggestions for Authors

This is an article that states that supplementation of specific metabolites can correct pathological alterations in cellular models of Kat6A . the cell model used is a dermal fibroblast derived from patients with KAT6A syndrome. While the idea is quite interesting, the data to support the very strong claims in the title ‘correction of pathological alterations in cellular models of KAT6A syndrome’. The claims made in writing are not supported by the data shown in the paper, both the conclusions from the data and the lack of biological replicates reported across the paper.

Major Points.

The introduction talks about a lot of items that are not relevant to their message. For example, the first paragraph is about ASD and finding a cure. Furthermore, the reference to autism as associated with ASD is wrong, KAT6A is only considered a class 2 risk gene and not directly causal to ASD.

 The abstract and Introduction section have been rewritten accordingly.

 Finally, references 9 and 15 are the same paper—its not clear by it is listed twice.

The reference has been corrected.

The reproduction should be rewritten to detail the background in the cellular metabolism and the hypothesis they are testing.

The abstract and Introduction sections have been rewritten.

 It is not clear why how the authors chose  the experimental design and metabolites for testing and what the rational for this.

The rationale of using cellular models and treatments have been included in the Introduction section.

 The title refers “pathological alterations” but its not outline what the authors are referring to here or in the introduction.

The specific pathological alterations have been included in the Introduction section.

Figure 1, western blots were not properly quantified and its hard to determine whether there was a significant effect for these. The claim of significant effect would require quantitative densitometry measurements of the blot, and testing the relative changes to control across at least 2-3 blots.

Western blots have been quantified across 3 blots. This information has been included in the Figure legends.

Figure 2, is completely uninterpretable. Its not clear what one is supposed to see and therefore any claims here are unjustified and not supported.

The images have been magnified for better visualization.

Figure 3: the western blot results observed for non-treated P3 different amont the same protein antibody between figure 1 and figure 3.  For example, in Figure 1, for  PDH and KDGH, this fibroblast samples has no expression of these enzymes. However in Figure 3, untreated condition this has quite visible and clear expression. This inconsistence is not explained in the literature.

New Western blots have been included in Figure 1 and 3, accordingly.

Figure 3 & 4 & 5, were only completed with one line and not with the other two. All data should be performed in all three lines. Its impossible to draw any concpusions from a comparison between 1 case and 1 control.  

Histone acetylation and bioenergetics assays of the two additional cell lines have been included in Supplementary Figures S10-S13.

 Please use more common notation for p-value assessment rather than the a- and aa notation which is quite confusion.

 For p-values notations, the symbol * has been used in all the graphs.

(page 3, line 101)

One of the main points of the paper suggest that there are new disease modifying therapies poposed from KAT6A. The paper cite is not a peer reviewed paper but an abstract from a small meeting and therefore hard to validate as a to whether a new treatment has truly been proposed.

The reference has been deleted

(Page 3, line 188)

The description of the RNA-analysis is really unclear and have some major analytic problems. First, there is not evidence that there were RNA-seq quality control metrics done. The cells were not mentioned to be tested for mycoplasma and this is quite common in patient-derived lines. The authors talk about RNA-extraction but they do nto give any information about the sequencing metrics (RNA library prep, QC on RNA, read length, coverage or whether the sequencing was paired end or single end). The lack of detail here makes one suspect on data quality. Finally, the authors state that use a p-value of 0.05. Is there any correction for multiple testing here? The lack of details on their RNA-seq process and pipeline further erode credibility on the data quality.

Information about quality control and sequencing metrics have been included. All cell lines were tested for mycoplasma. Details of RNA-seq process have been included in the Material and Methods section.

 Minor Points:

This paper includes three patients with three unique mutations. Two are truncating mutations and P2 is a missense mutations. The paper claims this is loss-of-function but there is not proof that this is in fact a true loss-of-function. The authors should dial back their claim on the effects of the mutation.  

The three cell lines showed low levels of mutant enzyme and reduced histone acetylation. These results suggest loss-of-function of the mutant enzymes.

 Reagents do not have any catalog numbers included and do not include the dilutions tested of the reagents making the work largely non-reproducible.

Reagents catalog numbers have been included accordingly.

  Typo in page2, line 86 “ very variable penetration” of signs and symptoms  of people with the same condition.

The typo has been corrected accordingly.

Reviewer 2 Report

Pantothenate and L‐carnitine supplementation corrects pathological alterations in cellular models of KAT6A syndrome

The manuscript by Munuera‐Cabeza et al.  aims to develop a model for KAT6A gene loss of function using human patient-derived cells. It identifies interesting changes in gene expression related to mitochondrial function and metabolism. Using a cellular stress assay, the authors argue that heterozygous KAT6A loss of function causes decreased cell viability which was rescued by CoA metabolism and mitochondrial activator drugs. While this could be a useful model for further elucidation of cellular mechanisms underlying KAT6A mutations, the manuscript in its current form has several issues.

My comments are below:

1.      Why were fibroblasts chosen as a model system to identify changes in in a neurodevelopmental disease? What was the rationale? This needs to be explained in detail to understand how the results from fibroblasts could recapitulate neurodevelopmental phenotypes.

2.      In the quantification of western blots, how many replicates of the blots were used for statistical analysis? It is important to indicate the statistical test in the figure legends. Without this information the western blot analysis is not clear.

3.      Quantification information, number of replicates and statistical test used need to be described in all figure legends with quantification data.

4.      How was mitochondrial mass analyzed and what were the results from that assay? How was no change in mitochondrial mass ascertained?

5.      “However, the expression levels of proteins involved in iron‐sulfur clusters biosynthesis such as ISCU (Iron‐sulfur cluster assembly enzyme), NFS1 (NFS1 cysteine desulfurase) and FXN (Frataxin) were not affected. In addition, intracellular iron accumulation assessed by Prussian blue staining was not observed in mutant KAT6A fibroblasts (Figure S3)”: What is the possible reason for this? Where are the blots/quantification for the proteins that did not change in levels?

6.       Imaging in Fig 2 is very hard to understand visually. Image quality is not good and it is unclear how viability changes in these representative images. A cell viability stain or better images are needed to show the difference between mutant and control lines.

7.      How were cells counted? Was a cell counter used? Were the cells counted manually? How many replicates/plates/batches of patient fibroblasts were analyzed to assay the cell viability?

8.      In Fig S4-6: Why was an endpoint of 72 hours selected to assess cell survival? The text makes reference to “growth rate” while the figures refer to “cell Proliferation”. It is unclear what these two parameters signify and how each of them is quantified.

All rescue experiments are conducted on a single patient line. Rationale for choosing this approach is needed. Rescue data from a single line is not sufficient to claim a mechanism

9.      In S7 there is a reference to panel C but no such panel exists. 

10.   Figures and figure legends need to be neater and more informative. For example, the labels in the heatmaps in Fig 7 are too small to read. Labels on the plots in Fig6 for example are illegible.

Author Response

REVIEWER 2

Pantothenate and L‐carnitine supplementation corrects pathological alterations in cellular models of KAT6A syndrome

The manuscript by Munuera‐Cabeza et al.  aims to develop a model for KAT6A gene loss of function using human patient-derived cells. It identifies interesting changes in gene expression related to mitochondrial function and metabolism. Using a cellular stress assay, the authors argue that heterozygous KAT6A loss of function causes decreased cell viability which was rescued by CoA metabolism and mitochondrial activator drugs. While this could be a useful model for further elucidation of cellular mechanisms underlying KAT6A mutations, the manuscript in its current form has several issues.

My comments are below:

  1. Why were fibroblasts chosen as a model system to identify changes in in a neurodevelopmental disease? What was the rationale? This needs to be explained in detail to understand how the results from fibroblasts could recapitulate neurodevelopmental phenotypes.

The rationale of using cellular models and treatments have been included in the Introduction section.

  1. In the quantification of western blots, how many replicates of the blots were used for statistical analysis? It is important to indicate the statistical test in the figure legends. Without this information the western blot analysis is not clear.

 The number of Western blotting replicates has been included in the Figure legends.

    3. Quantification information, number of replicates and statistical test       used need to be described in all figure legends with quantification data.

The number of Western blotting replicates has been included in the Figure legends.

  1. How was mitochondrial mass analyzed and what were the results from that assay? How was no change in mitochondrial mass ascertained?

VDAC expression levels were used as a marker of mitochondrial content.

 “In contrast, expression levels of VDAC1, a marker of mitochondrial content of cells [42], were not affected (Figure S3C and S3D). These results suggest that there is a down-regulation of several essential mitochondrial proteins in KAT6A fibroblasts. The decreased levels of essential mitochondrial proteins may lead to mitochondrial dysfunction, increased reactive oxygen species (ROS) production and reduced energy generation”.

 The expression levels of VDAC1 have been included in Supplementary Figure S3C and S3D.

 “However, the expression levels of proteins involved in iron-sulfur clusters biosynthesis such as ISCU (Iron-sulfur cluster assembly enzyme), NFS1 (NFS1 cysteine desulfurase) and FXN (Frataxin) were not affected. In addition, intracellular iron accumulation assessed by Prussian blue staining was not observed in mutant KAT6A fibroblasts (Figure S3)”: What is the possible reason for this? Where are the blots/quantification for the proteins that did not change in levels?

 The expression levels of proteins involved in iron-sulfur clusters biosynthesis such as ISCU (Iron-sulfur cluster assembly enzyme), NFS1 (NFS1 cysteine desulfurase) and FXN (Frataxin) have been included in Supplementary Figure S3C and S3D. 

  1. Imaging in Fig 2 is very hard to understand visually. Image quality is not good and it is unclear how viability changes in these representative images. A cell viability stain or better images are needed to show the difference between mutant and control lines.

The images in Figure 2 have been magnified for better visualization.

  1. How were cells counted? Was a cell counter used? Were the cells counted manually? How many replicates/plates/batches of patient fibroblasts were analyzed to assay the cell viability?

Cell viability was tested by live cell imaging counting and trypan blue 0.2% staining. Cell counting was acquired using the BioTekTM CytationTM 1 Cell Imaging Multi-Mode Reader (BioTek, Winooski, VT, United States).

All the assays were performed in triplicate. 

  1. In Fig S4-6: Why was an endpoint of 72 hours selected to assess cell survival? The text makes reference to “growth rate” while the figures refer to “cell Proliferation”. It is unclear what these two parameters signify and how each of them is quantified.

 The 72 hours endpoint was selected because cells showed a significant cell proliferation/death at this time.

 Cell survival was assessed by cell proliferation quantification. The text and figures legends have been rewritten accordingly.

All rescue experiments are conducted on a single patient line. Rationale for choosing this approach is needed. Rescue data from a single line is not sufficient to claim a mechanism

Histone acetylation and bioenergetics assays of the two additional cell lines (P2 and P3) have been included in supplementary figures S10-S13. 

  1. In S7 there is a reference to panel C but no such panel exists.

Panel C has been included accordingly. 

  1. Figures and figure legends need to be neater and more informative. For example, the labels in the heatmaps in Fig 7 are too small to read. Labels on the plots in Fig6 for example are illegible.

Labels have been corrected accordingly.

Round 2

Reviewer 1 Report

Major Points

For any of the claims to be considered as “correcting” a pathological defect, which is a claim that seems a bit of a stretch to have done this in a fibroblast cell line and with the few, low-sensitvity measures. This includes the RNAseq. While the authors have performed additional western blots for patients 2 and 3 and performed quantification, the RNAseq which would be the most convincing data . You cannot make conclusions on an n=1 sample. Please add RNA-seq data from P2 and P3, with and without the treatments and perform a more rigorous analysis.

Finally, the supplemental data does not include any tables or gene lists from the RNAseq data, only the gene ontology is represented which is contrary to the response saying they will include this data. 

The first sentence of the abstract doesn’t really make sense. Please make it read more clearly

The title needs to be toned down as I don’t believe the bar has been met to say that the treatment has “corrected the pathophysiological defect” in this syndrome.  Please change title, specifically the word “corrects” should be toned down to something like “decreases”, or diminishes because the effect of the mutation doesn’t completely go away in any of the data shown here.

Magnification of the figure does not make it more interpretable.

For all Figure legends, a proper title for the figure legend would help the reader understand what the message is.

For the figures, subpanels that integrate P1/2/3 rather than having them as separate or int he supplement would be more convincing.

Author Response

For any of the claims to be considered as “correcting” a pathological defect, which is a claim that seems a bit of a stretch to have done this in a fibroblast cell line and with the few, low-sensitvity measures. This includes the RNAseq. While the authors have performed additional western blots for patients 2 and 3 and performed quantification, the RNAseq which would be the most convincing data . You cannot make conclusions on an n=1 sample. Please add RNA-seq data from P2 and P3, with and without the treatments and perform a more rigorous analysis.

Protein expression levels, histone acetylation by immunofluorescence and bioenergetics assays have been performed in the three cell lines.

We have performed the Histone H3 Total Acetylation Colorimetric Detection Fast Kit, NAD+/NADH Assay Kit and RNAseq only in one cell line due to time and funding constraints.

Finally, the supplemental data does not include any tables or gene lists from the RNAseq data, only the gene ontology is represented which is contrary to the response saying they will include this data. 

The list of key genes from the RNAseq data has been included as supplementary file (RNAseq-data.xlsx), accordingly.

The first sentence of the abstract doesn’t really make sense. Please make it read more clearly

 We have changed the sentence accordingly

Mutations in several genes involved in the epigenetic regulation of gene expression have been considered risk alterations to different intellectual disability (ID) syndromes associated with autism spectrum disorder (ASD) features. Among them, it has been included the pathogenic variants of the lysine-acetyltransferase 6A (KAT6A) gene causing KAT6A syndrome.

 The title needs to be toned down as I don’t believe the bar has been met to say that the treatment has “corrected the pathophysiological defect” in this syndrome.  Please change title, specifically the word “corrects” should be toned down to something like “decreases”, or diminishes because the effect of the mutation doesn’t completely go away in any of the data shown here.

The title has been changed accordingly

Pantothenate and L-carnitine Supplementation Improves Pathological Alterations in Cellular Models of KAT6A Syndrome.

Magnification of the figure does not make it more interpretable.

Figure 2. Cell death in stress medium has been pointed out with white arrows. The quantification of cellular proliferation rate is shown in Supplementary Figure S4-6.

For all Figure legends, a proper title for the figure legend would help the reader understand what the message is.

The tittles of Figure legends have been changed, accordingly.

For the figures, subpanels that integrate P1/2/3 rather than having them as separate or int he supplement would be more convincing.

Given the amount of data, the integration of the tree cell lines in one figure is not feasible.

We have tried the reviewer`s suggestion but the result was that panels were smaller and difficult to visualize.

Reviewer 2 Report

Pantothenate and L‐carnitine supplementation corrects pathological alterations in cellular models of KAT6A syndrome

The manuscript by Munuera‐Cabeza et al.  aims to develop a model for KAT6A gene loss of function using human patient-derived cells. It identifies interesting changes in gene expression related to mitochondrial function and metabolism. Using a cellular stress assay, the authors argue that heterozygous KAT6A loss of function causes decreased cell viability which was rescued by CoA metabolism and mitochondrial activator drugs. 

The authors have addressed most of my questions and comments and this version is significantly better than the previous one. Additional data from more patient lines does provide more evidence to support the conclusions.

However, some data is partially complete. For example some analyses show data only from a 1 or 2 or combination of patient lines. The text should clarify why certain lines were chosen for certain experiments, or explain why data from all three patient lines is not available for those experiments.

The figures, especially fig 6 onwards still require more editing of the labels They are not legible.

Author Response

The manuscript by Munuera‐Cabeza et al.  aims to develop a model for KAT6A gene loss of function using human patient-derived cells. It identifies interesting changes in gene expression related to mitochondrial function and metabolism. Using a cellular stress assay, the authors argue that heterozygous KAT6A loss of function causes decreased cell viability which was rescued by CoA metabolism and mitochondrial activator drugs. 

The authors have addressed most of my questions and comments and this version is significantly better than the previous one. Additional data from more patient lines does provide more evidence to support the conclusions.

However, some data is partially complete. For example some analyses show data only from a 1 or 2 or combination of patient lines. The text should clarify why certain lines were chosen for certain experiments, or explain why data from all three patient lines is not available for those experiments.

We have performed the Histone H3 Total Acetylation Colorimetric Detection Fast Kit, NAD+/NADH Assay Kit and RNAseq only in one cell line due to time and funding constraints.

We have specified the cells lines used in the assays in the text and figure legends.

The figures, especially fig 6 onwards still require more editing of the labels They are not legible.

Labels of Figure 6 and 7 have been corrected, accordingly
